# Cholesteryl Ester Transfer Protein Genetic Variants Associated with Risk for Type 2 Diabetes and Diabetic Kidney Disease in Taiwanese Population

**DOI:** 10.3390/genes10100782

**Published:** 2019-10-08

**Authors:** Yu-Chuen Huang, Shih-Yin Chen, Shih-Ping Liu, Jane-Ming Lin, Hui-Ju Lin, Yu-Jie Lei, Yun-Chih Chung, Yu-Chi Chen, Yeh-Han Wang, Wen-Ling Liao, Fuu-Jen Tsai

**Affiliations:** 1School of Chinese Medicine, China Medical University, Taichung 404, Taiwan; yuchuen@mail.cmu.edu.tw (Y.-C.H.); chenshihy@mail.cmu.edu.tw (S.-Y.C.); d4301@seed.net.tw (J.-M.L.); irisluu2396@gmail.com (H.-J.L.); 2Department of Medical Research, China Medical University Hospital, Taichung 404, Taiwan; yjlei0824@gmail.com; 3Center for Translational Medicine, China Medical University Hospital, Taichung 404, Taiwan; spliu@mail.cmu.edu.tw; 4Graduate Institute of Biomedical Science, China Medical University, Taichung 404, Taiwan; 5Department of Ophthalmology, China Medical University Hospital, Taichung 404, Taiwan; 6Department of Public Health, China Medical University, Taichung 404, Taiwan; ai762928@gmail.com (Y.-C.C.); s7911235@gmail.com (Y.-C.C.); 7Department of Anatomical Pathology, Taipei Institute of Pathology, Taipei 103, Taiwan; yehanwang@gmail.com; 8Center for Personalized Medicine, China Medical University Hospital, Taichung 404, Taiwan; 9Graduate Institute of Integrated Medicine, China Medical University, Taichung 404, Taiwan; 10Department of Pediatric Internal Medicine, Children’s Hospital of China Medical University, Taichung 404, Taiwan; 11Department of Medical Genetics, China Medical University Hospital, Taichung 404, Taiwan

**Keywords:** CETP, HDL-C, type 2 diabetes, diabetic kidney disease, diabetic retinopathy

## Abstract

Cholesteryl ester transfer protein (CETP) plays an important role in lipid metabolism. Low levels of high-density lipoprotein cholesterol (HDL-C) increase the risk of type 2 diabetes (T2D). This study investigated CETP gene variants to assess the risk of T2D and specific complications of diabetic kidney disease (DKD) and diabetic retinopathy. Towards this, a total of 3023 Taiwanese individuals (1383 without T2D, 1640 with T2D) were enrolled in this study. T2D mice (*+Lepr^db^/+Lepr^db^*, *db/db*) were used to determine CETP expression in tissues. The A-alleles of rs3764261, rs4783961, and rs1800775 variants were found to be independently associated with 2.86, 1.71, and 0.91 mg/dL increase in HDL-C per allele, respectively. In addition, the A-allele of rs4783961 was significantly associated with a reduced T2D risk (odds ratio (OR), 0.82; 95% confidence interval (CI), 0.71–0.96)), and the A-allele of rs1800775 was significantly related to a lowered DKD risk (OR, 0.78; 95% CI, 0.64–0.96). CETP expression was significantly decreased in the T2D mice kidney compared to that in the control mice (T2D mice, 0.16 ± 0.01 vs. control mice, 0.21 ± 0.02; *p* = 0.02). These collective findings indicate that CETP variants in the promoter region may affect HDL-C levels. Taiwanese individuals possessing an allele associated with higher HDL-C levels had a lower risk of T2D and DKD.

## 1. Introduction

Type 2 diabetes (T2D) is a chronic metabolic disorder with a multifactorial pathogenesis. T2D involves complex interactions between environmental factors and genetic predisposition [1,2]. Macro- and microvascular diseases are devastating complications of diabetes and are the major causes of morbidity and early mortality in diabetic individuals [3]. Dyslipidemia is regarded as an important risk factor for cardiovascular disease due to its influence on atherosclerosis and is considered to be a major complication and a leading cause of mortality in T2D patients [4].

The characteristics of diabetic dyslipidemia include high serum triglyceride concentrations, low serum high-density lipoprotein cholesterol (HDL-C) levels, and elevated low-density lipoprotein cholesterol (LDL-C) levels. In several epidemiological studies, low levels of HDL-C were associated with an increased risk of T2D [5,6]. Factors reported determining HDL-C levels include hormonal, environmental, cultural, and regulatory genetic factors [7]. Cholesteryl ester transfer protein (CETP) is a protein that facilitates the exchange of cholesteryl esters between HDL-C and glyceride-rich lipoprotein [8]. CETP plays an important role in the metabolism of HDL-C. Genetic variations in CETP have been studied, with the variations being associated with reduced CETP activities as well as reduced risks of cardiovascular diseases [9,10,11,12,13,14]. However, whether CETP genetic variations impact on the risk of developing diabetes is controversial. Ethnic factors might play a role in the association in diabetes patients. Most CETP genetic association studies in Taiwanese populations have focused on Taq1B (rs708272) [15,16,17]. However, as the Taq1B polymorphism is located in the first intron of the CETP gene, it cannot clearly specify the effect of CETP variation on disease risk. In addition, only a few reports have examined the associations of CETP variants with microvascular complications in T2D patients.

In the present study, we evaluated the respective effects of three single nucleotide polymorphisms (SNPs) located in the CETP promoter region—rs3764261, rs4783961, and rs1800775—and a missense coding SNP (rs5882) of the CETP gene on the following: lipid concentrations, associated risk of T2D, and the specific complications of diabetic kidney disease (DKD) and diabetic retinopathy (DR) in a Taiwanese population. In addition, we used a T2D mouse model to determine the expression levels of CETP in adipose, kidney, and retinal tissues.

## 2. Methods

### 2.1. Study Participants

The study involved 1640 unrelated individuals with T2D (age range 24–96 years), who were recruited from the China Medical University Hospital (CMUH), Taichung, Taiwan. Individuals with type 1 diabetes (T1D), gestational diabetes, or maturity-onset diabetes of the young, were excluded. Subjects were diagnosed using the American Diabetic Association Criteria [18]. DKD was defined as the presence of micro—or macro-albuminuria and/or estimated glomerular filtration rate (eGFR) < 60 mL/min/1.73 m^2^ [19]. Of this group, eGFR values were obtained for 1325 of 1640 T2D subjects, and 367 of these 1325 T2D subjects were diagnosed with DKD. DR was diagnosed by expert ophthalmologists based on a complete ophthalmologic examination that included corrected visual acuity and fundoscopic examination; fundus photography was graded according to the American Academy of Ophthalmology proposed international scales for grading the severity of clinical DR [20]. Among the T2D group, 1063 subjects underwent ophthalmologic examination, and 475 of them were diagnosed with DR: 201 (18.9%) subjects were diagnosed with non-proliferative DR (NPDR), and 274 (25.8%) subjects were diagnosed with proliferative DR (PDR). A questionnaire was prepared to collect information regarding sex, age, and age at time of diagnosis of diabetes. For each patient, systolic and diastolic blood pressure, waist and hip circumferences, body mass index (BMI), and glycated hemoglobin A_1C_ (HbA_1C_), total cholesterol, HDL-C, LDL-C, triglycerides, and eGFR levels were determined. The non-diabetic controls consisted of 1383 subjects (age range 30–70 years) who were randomly selected from the Taiwan Biobank (TWB) [21], a national large-scale data source with genetic and demographic data from Taiwanese individuals between the ages of 30 and 70 years (accessed at https://www.twbiobank.org.tw/new_web_en/index.php). The inclusion criteria for selecting controls in the association study were no history of T1D, diagnosis of T2D, and an HbA_1C_ < 6.0%. Among these 1383 non-diabetic controls, 349 individuals aged ≥55 years and 253 individuals residing in Taichung and neighboring countries were selected as an age control group and a geographical region control group, respectively, to perform the association studies. The study was approved by the institutional review boards on CMUH, Taiwan (IRB No. CMUH103-REC2-071(CR-3)), and by the Ethics and Governance Council (EGC) of Taiwan Biobank, Taiwan. The information of non-diabetic controls was applied from Taiwan Biobank which is governed by the EGC and the Ministry of Health and Welfare, Taiwan. The individual genotype information could not be released due to the regulations but the summary statistics could be queried in the Taiwan View website (https://taiwanview.twbiobank.org.tw/index).

### 2.2. SNP Selection and Genotyping

Three SNPs—rs3764261 (-2568A/C), rs4783961 (-998A/C), and rs1800775 (-629A/C)—located in the promoter region (positions: chr16: 56959412, 56960982, 56961324 bp, respectively), as well as a missense coding SNP rs5882 (+16A/G) in exon 14 (position: chr16, 56982180 bp) of the CETP gene were included. The SNPs were selected from the human genome database (accessed at https://www.ncbi.nlm.nih.gov/genome/guide/human/) and, based on previously published reports, have been associated with an effect on HDL-C levels [9,10,22,23]. SNPs were excluded if their minor allele frequencies in the HapMap CHB population were <0.01, or call rates were <98%. Additionally, SNPs that deviated from the Hardy–Weinberg equilibrium (HWE) were excluded (*p* < 0.05 indicated deviance from HWE). T2D subjects were genotyped using the Hap550K-BeadChip (Illumina, San Diego, CA, USA), which has been used previously for genome-wide association studies in the Han Chinese population of Taiwan [24]. For the non-diabetic controls from TWB, DNA was isolated from blood samples using a Chemagic™ 360 instrument following the manufacturer’s instructions (PerkinElmer, Waltham, MA, USA). SNP genotyping used custom-designed 653K TWB chips and was conducted on the Axiom Genome-Wide Array Plate System (Affymetrix, Santa Clara, CA, USA) [21].

### 2.3. T2D Mouse Model

Six weeks-old T2D mice (BKS.Cg-*Dock7^m^*+/+*Lepr^db^*/JNarl, abbreviation *db/db*), and their non-diabetic littermates (control mice, abbreviation *+/+*) were obtained from the National Laboratory Animal Center (Taipei, Taiwan). Six male mice per group aged 32 weeks were used for the experiment. All mice were housed under light/dark conditions (12 h/12 h) with free access to water and food. Tail vein blood samples were used to monitor blood glucose levels every two weeks (Roche, Mannheim, Germany). All animal care and handling were approved by the Institutional Animal Care and Use Committee of China Medical University (CMUIACUC-2017-207-1).

### 2.4. Western Blot

Radioimmunoprecipitation lysis buffer (Sigma-Aldrich, St. Louis, MO, USA) containing protease inhibitors and phosphatase inhibitors (Roche, Indianapolis, IN, USA) was used to extract the proteins from mouse kidney and retina tissues. Protein extracts (20 μg) were separated using 12% (*w*/*v*) sodium dodecyl sulfate-polyacrylamide gel and then transferred to 0.45 μm pore size nitrocellulose membranes (Millipore, Billerica, MA, USA). The membranes were incubated with anti-CETP primary antibody (dilution 1:500; GeneTex, Austin, TX, USA) overnight at 4 °C, followed by incubation with horseradish peroxidase-conjugated secondary antibody (GeneTex) at room temperature for 1 h. Anti-β-actin (dilution 1:6000, Novus Biological, Littleton, CO, USA) was used as an internal control. The blot signals were developed using an enhanced chemiluminescence system (ChemiGenius XE Bio Imaging System; Syngene, Frederick, MD, USA). The ImageJ program (NIH, Bethesda, MD, USA) was used to quantify protein expression, which was normalized to the expression of the internal control.

### 2.5. Immunohistochemistry

Mouse adipocyte tissue sections (5-μm) were deparaffinized and soaked in a 3% hydrogen peroxide solution in distilled water for 5 min to counteract endogenous peroxidase reactions. The sections were further incubated with primary antibody against CETP (dilution 1:100, GeneTex), followed by incubation with horseradish peroxidase-conjugated secondary antibody. For all sections, the presence of peroxidase was detected by the addition of 3, 3′-diaminobenzidine tetrahydrochloride solution, and finally counterstained with hematoxylin.

### 2.6. Statistical Analyses

The Student’s *t*-test for continuous variables and chi-square test for categorical variables were used to compare the characteristics and clinical data of T2D subjects and controls. One-way analysis of variance (ANOVA) or general linear regression models were used to analyze the effect of the *CETP* gene SNP genotypes on lipid levels. Chi-square goodness of fit test was used to check compliance with the Hardy–Weinberg equilibrium. Association analysis was carried out to compare genotype distribution between T2D subjects and non-diabetic controls, or subjects with or without DKD, or subjects with or without DR, using additive models. Odds ratios (ORs) and 95% confidence intervals (CIs) were determined by logistic regression and were adjusted for age, sex, HbA_1C_, or BMI levels. All statistical analyses were performed using the IBM SPSS Statistics 22 (IBM Co., Armonk, NY, USA). A *p*-value of less than 0.05 was considered statistically significant. Haploview software (version 4.2) was used to estimate the frequencies of CETP haplotypes and to calculate the linkage disequilibrium (D’) between any two loci [25].

## 3. Results

The study included 1383 non-diabetic controls and 1640 patients with T2D. The demographic and clinical characteristics of these groups are summarized in Table 1. In brief, 48.1% of the non-diabetic controls and 50.8% of the T2D patients were men. Non-diabetic controls were significantly younger in age at the time of the study and displayed lower HbA_1C_ levels, lower BMI levels, lower systolic and diastolic blood pressure, lower waist–hip ratio, higher total cholesterol, higher HDL-C, lower LDL-C, lower triglycerides, and higher eGFR levels than did the T2D patients. Among the T2D patients, the mean duration of diabetes was 11.0 ± 8.7 years and 475 of the 1063 T2D patients were diagnosed with DR (18.9% with NPDR and 25.8% with PDR); further, of the 1325 of the 1640 T2D subjects for whom eGFR values were obtained, 27.7% (367/1325) had DKD.

Table 2 summarizes the genotype distribution and lipoprotein levels in the different genotypes of CETP SNPs, including rs3764261, rs4783961, rs1800775, and rs5882 among the non-diabetic controls. The A-allele of rs3764261 and rs4783961, but not the A-allele of rs1800775 and the G-allele of rs5882, were significantly associated with higher HDL-C levels. Nevertheless, none of these CETP SNPs were associated with total cholesterol, triglyceride, and LDL-C levels.

We estimated the change in HDL-C in the different genotypes of CETP SNPs. The A-alleles of rs3764261, rs4783961, and rs1800775 were independent of age and sex and were associated with a 2.86, 1.71, and 0.91 mg/dL increase in the levels of HDL-C per allele, respectively (Table 3). We did not observe a significant association between the change in HDL-C levels and rs5882, and between the other lipids, including total cholesterol, triglycerides, and LDL-C and these four CETP SNPs.

We then analyzed the risk of CETP variants in patients with T2D and non-diabetic controls. As shown in Table 4, only the A-allele of rs4783961 was associated with a significantly decreased risk of T2D in the additive (per allele) model after adjusting for sex and age (adjusted odds ratio (OR), 0.82; 95% confidence interval (CI), 0.71–0.96) and adjusting for sex, age, and BMI (adjusted OR, 0.81; 95% CI, 0.70–0.95). In addition, we compared the risk of CETP variants between T2D patients and non-diabetic controls with age ≥55 years (age control group) or non-diabetic controls residing in regions similar to those of T2D patients (geographical region control). As shown in Table 4, the A-allele of rs4783961 was associated with a significantly decreased risk of T2D in the model after adjusting for sex and age and for sex, age, and BMI in both the non-diabetic controls with age ≥55 years and non-diabetic geographical region controls.

We subsequently compared the risk of CETP variants in the T2D patients according to the presence and absence of DKD (eGFR < 60 and ≥ 60, respectively) and with/without DR. As shown in Table 5, the A-allele of rs1800775 was independently associated with sex and age, and HbA_1C_ was associated with decreased risk of T2D with DKD in the additive model (adjusted OR, 0.78; 95% CI, 0.64–0.96). The A-allele of rs1800775 was also independently associated with sex, age, HbA_1C_, and BMI was associated with decreased risk of T2D with DKD in the additive model (adjusted OR, 0.80; 95% CI, 0.65–0.98). However, none of the studied CETP variants was associated with the risk of T2D in patients with DR.

Pairwise linkage disequilibrium analysis revealed a strong linkage disequilibrium in three of four SNPs (rs3764261, rs4783961, and rs1800775) located in the CETP promoter region (D’ ≥ 0.945). As shown in Table 6, haplotypes AAA and CAA were associated with a significantly decreased risk of T2D (OR, 0.83; 95% CI, 0.72–0.97 and OR, 0.73; CI, 0.57–0.92, respectively) compared with haplotype CGC.

The T2D mouse model was used to evaluate CETP expression in adipose, kidney, and retinal tissues of 32-week-old mice (Figure 1). Western blots showed that CETP levels were significantly decreased in the kidney tissue of T2D mice compared to those in non-diabetic control mice (relative CETP expression: T2D mice, 0.16 ± 0.01 vs. control mice, 0.21 ± 0.02; *p* = 0.020, Figure 1a). No significant changes were observed in CETP expression in the retina between T2D and control mice (relative CETP expression: T2D mice, 0.63 ± 0.12 vs. control mice, 0.72 ± 0.04; *p* = 0.467, Figure 1b). Immunohistochemical staining showed that T2D mice had larger adipocytes and inflammatory cells (indicated by arrowhead and arrow in Figure 2, respectively) than did control mice at 32 weeks of age. CETP expression was predominant in the adipocyte membrane in T2D mice compared to that in control mice.

## 4. Discussion

In the present study, three SNPs located in the promoter region were observed to be independent of age and sex and were significantly associated with HDL-C levels. In particular, SNP rs3764261 was associated with an HDL-C increase of 2.86 mg/dL per A-allele (*p* = 1.00 × 10^−6^), whereas A-alleles of rs4783961 and rs1800775 were associated with HDL-C increases of 1.71 mg/dL and 0.91 mg/dL per A-allele, respectively, at nominal significance level (*p* = 0.001 and 0.045, respectively), in the non-diabetic controls. We also observed that at a nominal significance level, the A-allele of rs4783961 was associated with a decreased risk of T2D (OR, 0.82; 95% CI, 0.71–0.96) and the A-allele of rs1800775 was associated a reduced risk of DKD (OR, 0.78; 95% CI, 0.64–0.96). However, these results were not adjusted for multiple testing, and thus should be confirmed in other larger, independent studies. HDL-C may stimulate pancreatic β-cell insulin secretion, modulate glucose uptake in skeletal muscle, and contribute to the pathophysiology of T2D [26]. Low levels of HDL-C have been consistently associated with an increased risk of T2D [5,6]. However, the genetic-related life-long reduction in HDL-C levels is not associated with increased T2D risk in the general population [23]. In addition, low levels of HDL-C were reported as an independent risk factor for the development of DKD in a large diabetic population [27]. Furthermore, the four CETP SNPs that were presently analyzed were not associated with DR risk. These results are consistent with a previous report that did not find evidence for a causal role of the lipid fractions in DR [28]. Nevertheless, in a 9-year follow-up study, CETP Taq1B polymorphism was reported to be associated with the development of DR in women with T2D [29].

There are several limitations to the present study. First, we did not measure CETP activity, and thus it is unclear whether the CETP SNPs affect CETP activity and whether they are associated with the risk of T2D or DKD. CETP promoter SNPs rs3764261 and rs4783961 were also investigated previously [22]. The authors found no association between the level of CETP activity and CETP polymorphisms in an Asian Indian diabetic cohort comprising 2431 subjects [22]. In addition, the authors reported that CETP activity did not differ between T2D patients and normoglycemic controls [22]. However, CETP activity was positively correlated with HDL-C levels [22]. Moreover, another report indicated that CETP activity was elevated in T1D patients with DKD, but was not responsible for the lowered HDL-C levels [30]. Another report also indicated no significant association between CETP genetic polymorphisms and DKD in T1D [31]. The reason for the inconsistency between our findings on the association between CETP polymorphisms and DKD risk in T2D patients and those of previous studies is unknown. Thus, further studies are needed to determine whether CETP plays different roles during the development of T1D and T2D. Further, whether increased or decreased CETP activity is favorable for the development of diabetes needs to be investigated. We found that three CETP variants located in the promoter region were strongly correlated with HDL-C levels, even though none of the CETP SNPs resulted in a functional change in CETP. It is conceivable that the CETP promoter region may harbor an as-yet to be discovered functional variant, which is in strong linkage disequilibrium with these SNPs. Second, the major limitation of this study is the selection of non-diabetic controls. Non-diabetic controls were randomly selected from the general Taiwan population from the Taiwan Biobank. However, the T2D patients were recruited from a local region, specifically from the CMUH of Taichung, Taiwan. In addition, the non-diabetic control subjects were significantly younger than the T2D patients and had significantly lower BMI. Therefore, we selected two other non-diabetic control groups, namely age control (age ≥ 55 years) group and geographical region control group, to perform the association studies (Table 4 and Table 5). Logistic regression analyses for comparing the age control and region control groups revealed that the T2D risk was similar between the two groups compared with that in the entire non-diabetic control group. Although the sample size was not very large, we believe that the risk obtained for the comparison between T2D patients and non-diabetic controls in our study is valid. Furthermore, we also added BMI as a covariate to perform logistic regression analyses to assess the T2D risk and DKD risk (Table 4 and Table 5). The OR for the T2D risk was similar among the covariates only after adjusting for age and sex, and the DKD risk was similar among the covariates only after adjusting for age, sex, and HbA_1C_. We believe that the difference in BMI between T2D patients and non-diabetic controls was eliminated from the analysis models. Third, the gene–diet interaction between CETP genetic polymorphisms and dietary fat intake could have an effect on plasma HDL-C level [32], and because the present study did not measure dietary fat intake levels in T2D patients and non-diabetic controls, dietary fat intake could be a potential confounder.

CETP expression in the T2D mouse model was decreased in the kidney and retinal tissues compared with that in control mice but was significantly decreased only in kidney tissue. We previously reported that db/db T2D mice exhibited features of the early clinical stages of DR at 32 weeks of age [33]. Here, we did not observe a significant change in the retinal CETP expression between T2D mice with DR and control mice. In terms of adipose tissue, it seems that CETP expression was increased in the T2D mice, particularly in the adipocyte membrane. It is possible that CETP has a diverse expression in different tissues and may play different roles in different organs. Indeed, mice lack CETP activity in plasma [34] and are, therefore, not good models for studies involving CETP, cholesterol, and related pathologies. Further studies are warranted to investigate this issue. The previous studies using human CETP minigene-expressing transgenic mice that were crossed with db/db strain suggested an antiatherogenic effect of CETP in the context of diabetic obesity [35]. Therefore, CETP expression may play an important role in the prevention of atherogenic lipoprotein profiles and atherosclerosis in diabetic mice [35].

## 5. Conclusions

In conclusion, CETP variants in the promoter region may affect HDL-C levels. Taiwanese individuals possessing an allele associated with higher HDL-C may have lower risks of T2D and DKD.

## Figures and Tables

**Figure 1 genes-10-00782-f001:**
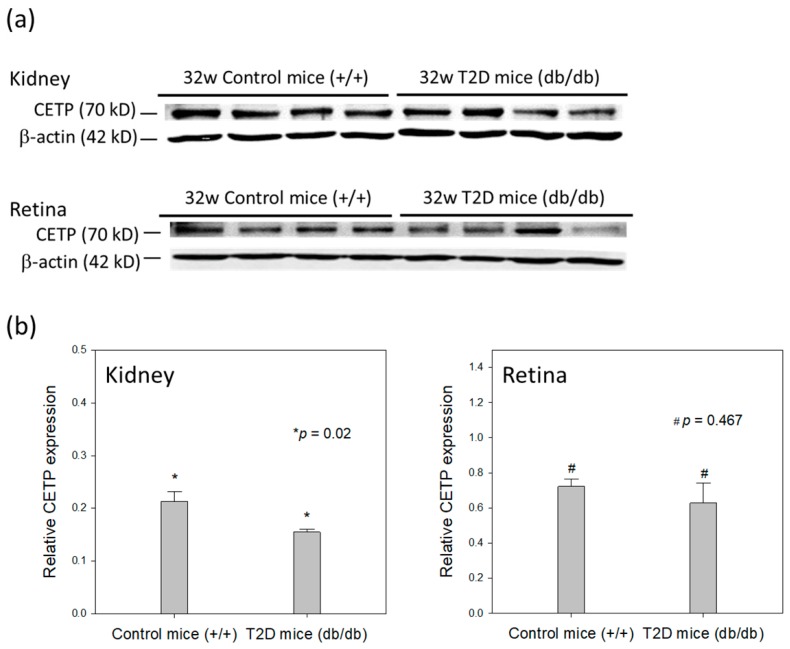
(**a**) Representative Western blot image of CETP expression in T2D and control mouse kidney and retina tissues. (**b**) CETP expression relative to that of β-actin in mouse kidney and retina. Data are presented as mean ± SD. * *p* = 0.020 (*t*-test); # *p* = 0.467 (*t*-test).

**Figure 2 genes-10-00782-f002:**
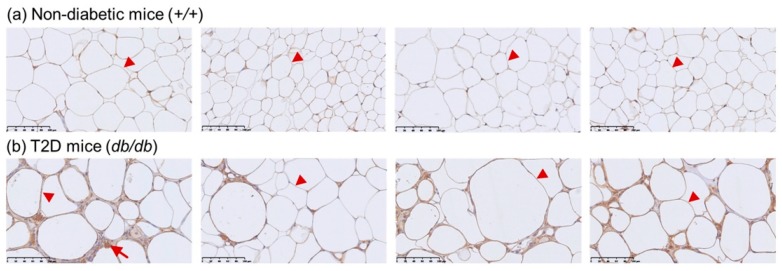
Representative images of immunohistochemical staining of CETP expression in mouse adipose tissue at 32 weeks of age (magnification 400×) (**a**) Non-diabetic mice (the arrowhead indicates adipocytes). (**b**) CETP is prominently expressed in the membrane of adipocyte in the T2D mouse model (arrowhead indicates adipocyte; arrow indicates cell with signs of inflammation).

**Table 1 genes-10-00782-t001:** Characteristics and clinical profiles of the study subjects.

	Non-Diabetic Control N = 1383	T2D Patients N = 1640	*p* Value
Sex			
*Male*	48.1%	50.8%	0.138
*Female*	51.9%	49.2%	
Age at study (mean ± SD, years)	46.0 ± 10.6	62.0 ± 11.3	<0.001
Duration of diabetes (mean ± SD, years)	--	11.0 ± 8.7	--
HbA_1C_ (%)	5.5 ± 0.3	7.9 ± 1.6	<0.001
Body mass index (kg/m^2^)	23.9 ± 3.5	25.4 ± 4.0	<0.001
Systolic blood pressure (mean ± SD, mmHg)	112.9 ± 16.6	140.4 ± 19.5	<0.001
Diastolic blood pressure (mean ± SD, mmHg)	71.2 ± 11.4	79.4 ± 12.1	<0.001
Waist–hip ratio (mean ± SD)	0.87± 0.23	0.93 ± 0.07	<0.001
Total cholesterol (mean ± SD, mg/dL)	190.9 ± 34.9	185.4 ± 41.8	<0.001
HDL-cholesterol (mean ± SD, mg/dL)	54.8 ± 12.9	48.3 ± 14.9	<0.001
LDL-cholesterol (mean ± SD, mg/dL)	119.7 ± 31.9	115.1 ± 37.0	0.001
Triglycerides (mean ± SD, mg/dL)	106.8 ± 67.2	166.0 ± 131.5	<0.001
eGFR (mean ± SD, mL/min/1.73 m^2^) ^1^ *≥60* *<60 (diabetic kidney disease, DKD)*	109.8 ± 26.0 1371 (99.1%) 12 (0.9%)	83.8 ± 40.6 958 (72.3%) 367 (27.7%)	<0.001 <0.001
Diabetic retinopathy severity scales ^2,3^			
*Non-DR*	--	588 (55.3%)	--
*Non-proliferative DR*	--	201 (18.9%)	--
*Proliferative DR*	--	274 (25.8%)	--

T2D: type 2 diabetes; SD: standard deviation; HbA_1C_: hemoglobin A_1C_; eGFR: estimated Glomerular filtration rate; ^1^ eGFR values were obtained for 1325 of 1640 T2D subjects; ^2^ According to the American Academy of Ophthalmology proposed international scales for severity of clinical diabetic retinopathy; ^3^ 1063 subjects underwent the ophthalmologic examinations.

**Table 2 genes-10-00782-t002:** Lipoprotein level in different genotypes of single nucleotide polymorphisms (SNPs) in the cholesteryl ester transfer protein (CETP) gene among non-diabetic controls.

dbSNP ID/Physical Position (bp)	Genotype	Non-Diabetic Controls N = 1383	Total Cholesterol (mg/dL)	*p* Value	Triglycerides (mg/dL)	*p* Value	HDL-Cholesterol (mg/dL)	*p* Value	LDL-Cholesterol (mg/dL)	*p* Value
rs3764261/	AA	51 (3.7)	189.5 ± 31.9	0.759	107.4 ± 57.8	0.910	58.9 ± 12.9	1.34 × 10^−4^	114.2 ± 30.6	0.421
56959412	AC	357 (25.9)	192.0 ± 34.8		105.5 ± 61.3		56.7 ± 13.4		119.3 ± 32.2	
	CC	971 (70.4)	190.5 ± 35.2		107.3 ± 69.8		53.9 ± 12.7		120.1 ± 31.9	
rs4783961/	AA	91 (6.7)	194.0 ± 34.6	0.603	109.6 ± 58.1	0.736	57.6 ± 12.5	0.011	119.7 ± 32.4	0.829
56960982	AG	457 (33.4)	191.6 ± 34.0		107.7 ± 77.3		55.7 ± 13.1		119.1 ± 31.7	
	GG	820 (59.9)	190.5 ± 35.5		105.3 ± 60.6		54.1 ± 12.8		120.2 ± 32.1	
rs1800775/	AA	369 (26.7)	191.7 ± 35.8	0.328	103.3 ± 53.3	0.499	56.0 ± 13.0	0.094	119.7 ± 33.3	0.198
56961324	AC	686 (49.7)	189.6 ± 35.3		108.4 ± 74.8		54.3 ± 13.0		118.5 ± 31.9	
	CC	326 (23.6)	192.9 ± 3.0		107.2 ± 64.4		54.3 ± 12.7		122.4 ± 29.9	
rs5882/	AA	400 (28.9)	191.7 ± 32.6	0.394	111.9 ± 78.8	0.193	54.0 ± 12.7	0.264	121.2 ± 29.6	0.330
56982180	AG	698 (50.5)	189.7 ± 36.0		104.9 ± 63.3		54.9 ± 13.1		118.5 ± 32.8	
	GG	284 (20.5)	192.7 ± 35.5		104.4 ± 57.8		55.6 ± 12.8		120.8 ± 32.7	

**Table 3 genes-10-00782-t003:** Association between CETP genetic variants and lipid levels in non-diabetic controls.

	Effect/Other Allele ^1^	N	Per Effect Allele (mg/dL)	SE	*p* Value ^2^
HDL-cholesterol					
rs3764261	A/C	1379	2.86	0.59	1.00 × 10^−6^
rs4783961	A/G	1368	1.71	0.52	0.001
rs1800775	A/C	1381	0.91	0.46	0.045
rs5882	G/A	1382	0.56	0.65	0.231
LDL-cholesterol					
rs3764261	A/C	1379	−2.24	1.55	0.150
rs4783961	A/G	1368	−1.11	1.37	0.417
rs1800775	A/C	1381	−1.44	1.19	0.227
rs5882	G/A	1382	−0.53	1.21	0.660
Total cholesterol					
rs3764261	A/C	1379	−0.03	1.68	0.988
rs4783961	A/G	1368	0.84	1.49	0.571
rs1800775	A/C	1381	−0.73	1.29	0.572
rs5882	G/A	1382	−0.12	1.31	0.928
Triglycerides					
rs3764261	A/C	1379	−2.09	3.21	0.515
rs4783961	A/G	1368	1.76	2.80	0.529
rs1800775	A/C	1381	−2.23	2.46	0.364
rs5882	G/A	1382	−3.41	2.49	0.172

^1^ Effect allele is defined as the allele associated with higher HDL-cholesterol; ^2^ Adjusted for age and sex.

**Table 4 genes-10-00782-t004:** Genotypic distribution of SNPs in the CETP gene and associated with risk for type 2 diabetics.

dbSNP ID/Effect Allele ^1^	Genotype/Effect Allele Number	T2D Patients N = 1640	Non-Diabetic Controls N = 1383	T2D vs. Non-Diabetic Controls OR (95% CI) *p*-Value	Non-Diabetic Controls (Age ≥ 55 Years) N = 349	T2D vs. Non-Diabetic Controls (Age ≥ 55 Years) OR (95% CI) *p*-Value	Non-Diabetic Controls (Geographical Region Control) ^2^ N = 253	T2D vs. Non-Diabetic Controls (Geographical Region Control) (95% CI) *p*-Value
rs3764261/A	CC/0	1104 (70.9)	971 (70.4)	0.91 (0.77–1.08) 0.270 ^3^ 0.90 (0.76–1.07) 0.231 ^4^	243 (69.6)	0.90 (0.72–1.12) 0.334 ^3^ 0.90 (0.72–1.12) 0.340 ^4^	168 (66.9)	0.88 (0.66–1.16) 0.352 ^3^ 0.87 (0.65–1.15) 0.322 ^4^
	AC/1	420 (27.0)	357 (25.9)		92 (26.4)		73 (29.1)	
	AA/2	34 (2.2)	51 (3.7)		14 (4.0)		10 (4.0)	
rs4783961/A	GG/0	970 (62.3)	820 (59.9)	0.82 (0.71–0.96) 0.011 ^3^ 0.81 (0.70–0.95) 0.007 ^4^	200 (57.3)	0.80 (0.66–0.97) 0.021 ^3^ 0.79 (0.65–0.96) 0.016 ^4^	138 (54.8)	0.76 (0.59–0.97) 0.027 ^3^ 0.74 (0.57–0.95) 0.020 ^4^
	AG/1	523 (33.6)	457 (33.4)		124 (35.5)		96 (38.1)	
	AA/2	65 (4.2)	91 (6.7)		25 (7.2)		18 (7.1)	
rs1800775/A	CC/0	375 (24.1)	326 (23.6)	0.94 (0.83–1.07) 0.339 ^3^ 0.93 (0.82–1.06) 0.272 ^4^	81 (23.2)	0.94 (0.79–1.11) 0.448 ^3^ 0.93 (0.79–1.10) 0.411 ^4^	58 (22.9)	0.95 (0.77–1.18) 0.648 ^3^ 0.94 (0.75–1.17) 0.563 ^4^
	AC/1	790 (50.7)	686 (49.7)		172 (49.3)		126 (49.8)	
	AA/2	393 (25.2)	369 (26.7)		96 (27.5)		69 (27.3)	
rs5882/G	AA/0	474 (30.5)	400 (28.9)	0.94 (0.83–1.07) 0.379 ^3^ 0.93 (0.82–1.06) 0.269 ^4^	100 (28.7)	0.97 (0.82–1.14) 0.693 ^3^ 0.95 (0.81–1.13) 0.580 ^4^	74 (29.2)	1.03 (0.83–1.28) 0.786 ^3^ 1.00 (0.80–1.25) 0.975 ^4^
	AG/1	760 (48.9)	698 (50.5)		178 (51.0)		131 (51.8)	
	GG/2	319 (20.5)	284 (20.5)		71 (20.3)		48 (19.0)	

dbSNP ID, SNP database identification; T2D, type 2 diabetes; DKD, diabetic kidney disease; DR, diabetic retinopathy; OR, odds ratio; CI, confidence interval; ^1^ Effect allele is defined as the allele associated with higher HDL-cholesterol; ^2^ geographical region controls: non-diabetic controls residing in Taichung and neighboring countries; ^3^ Adjusted for age, and sex; ^4^ Adjusted for age, sex, and BMI.

**Table 5 genes-10-00782-t005:** Genotypic distribution of SNPs in CETP gene and associated with risk for type 2 diabetes (T2D) patients with diabetic kidney disease (DKD) and diabetic retinopathy (DR).

dbSNP ID/Effect Allele ^1^	Genotype/Effect Allele Number	T2D Patients with DKD/without DKD N = 367/958	T2D with DKD vs. without DKD OR (95% CI), *p*-Value	T2D Patients With DR/ Non-DR N = 475/588	DR vs. Non-DR OR (95% CI), *p*-Value
rs3764261/A	CC/0	235 (74.4)/650 (70.0)	0.78 (0.58–1.04), 0.094 ^2^	335 (70.5)/406 (69.2)	0.90 (0.70–1.16), 0.408 ^2^
	AC/1	74 (23.4)/254 (27.4)	0.80 (0.60–1.08), 0.154 ^3^	133 (28.0)/169 (28.8)	0.90 (0.70–1.16), 0.414 ^3^
	AA/2	7 (2.2)/24 (2.6)		7 (1.5)/12 (2.0)	
rs4783961/A	GG/0	207 (65.5)/574 (61.9)	0.82 (0.63–1.06), 0.126 ^2^	295 (62.1)/360 (61.2)	0.90 (0.72–1.12), 0.341 ^2^
	AG/1	97 (30.7)/310 (33.4)	0.84 (0.65–1.09), 0.186 ^3^	164 (34.5)/202 (34.4)	0.89 (0.71–1.12), 0.318 ^3^
	AA/2	12 (3.8)/44 (4.7)		16 (3.4)/26 (4.4)	
rs1800775/A	CC/0	85 (26.9)/221 (23.8)	0.78 (0.64–0.96), 0.019 ^2^	104 (21.9)/142 (24.1)	0.98 (0.82–1.17), 0.816 ^2^
	AC/1	169 (53.5)/456 (49.1)	0.80 (0.65–0.98), 0.032 ^3^	251 (53.0)/290 (49.3)	0.98 (0.82–1.17), 0.810 ^3^
	AA/2	62 (19.6)/252 (27.1)		119 (25.1)/156 (26.5)	
rs5882/G	AA/0	105 (33.4)/270 (29.2)	0.83 (0.68–1.02), 0.083 ^2^	144 (30.4)/175 (29.9)	0.94 (0.79–1.13), 0.506 ^2^
	AG/1	155 (49.4)/458 (49.5)	0.83 (0.67–1.03), 0.086 ^3^	237 (50.0)/277 (47.4)	0.94 (0.79–1.13), 0.506 ^3^
	GG/2	54 (17.2)/197 (21.3)		93 (19.6)/133 (22.7)	

dbSNP ID, SNP database identification; T2D, type 2 diabetes; DKD, diabetic kidney disease; DR, diabetic retinopathy; OR, odds ratio; CI, confidence interval; ^1^ Effect allele is defined as the allele associated with higher HDL-cholesterol; ^2^ Adjusted for age, sex, and HbA_1C_; ^3^ Adjusted for age, sex, HbA_1C_, and BMI.

**Table 6 genes-10-00782-t006:** Haplotype analysis of gene encoding *CETP.*

Haplotype *rs3764261/rs4783961/rs1800775*	Non-Diabetic Controls	T2D Patients	T2D vs. Non-Diabetic Controls	*p*-Value
	N (%)	N (%)	OR (95% CI)	
CGC	1322 (48.2)	1686 (51.7)	1.00 (ref)	
CGA	804 (29.3)	939 (28.8)	0.92 (0.81–1.03)	0.146
AAA	458 (16.7)	487 (14.9)	0.83 (0.72–0.97)	0.015
CAA	161 (5.9)	149 (4.6)	0.73 (0.57–0.92)	0.007

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
