# Peer review of "Cholesteryl Ester Transfer Protein Genetic Variants Associated with Risk for Type 2 Diabetes and Diabetic Kidney Disease in Taiwanese Population"

_genes, 2019, doi:10.3390/genes10100782_

Round 1

Reviewer 1 Report

Well written and interesting manuscript.

Reviewer 2 Report

There are two major flows in this work, in my view.

i) The non-diabetic control group has a significantly lower mean age compared to the T2D group, which means that one cannot exclude the possibility that some of the control group participants could develop T2D at an older age (e.g., close to the mean age of the T2D group). That, in turn, means that while comparisons within the T2D group (DKD/non-DKD; DR/non-DR) remain valid, the validity of the comparisons between controls and T2D patients is compromised. This is a shame because 1383 controls require a great deal of genotyping, but the authors should have exercised greater caution in their selection.

ii) To the best of my knowledge, mice are not good models for studies involving CETP, cholesterol, and related pathologies because (unless genetically engineered to do so) mice do not express CETP in the liver. The authors are probably aware of since they have only presented CETP expression data from the kidneys, retina and adipose tissue, but they made no effort to associate their use of the mouse model with functions that CETP may have in those tissues, different from its well documented participation in reverse cholesterol transport, in humans and in other animals.

Minor comments

The authors could have presented some information concerning the possible linkage between the four SNPs (if any).

Some confusion is apparent concerning the use of statistical terms such as correlation, association, the role and use of covariates, etc.

Reviewer 3 Report

In this manuscript, the authors examined 4 common gene variants of human CETP gene, i.e., 3 promoter region single nucleotide polymorphisms (SNPs) [rs3764261 (-2568 C>A), rs4783961 (-998 G>A), rs1800775 (-629 A>C)] and 1 coding region non-synonymous SNP rs5882 (exon 14 +16A>G) to assess their effects on risks of type 2 diabetes (T2D), and specific T2D complications, i.e., diabetic kidney disease (DKD) and diabetic retinopathy (DR). Based on a case control design, 1383 non-diabetic controls and 1640 T2D patients in a Taiwanese population were studied. Further, the authors used a T2D mouse model (+Leprdb/+Leprdb, db/db) as well as non-diabetic mice (+/+) to compare the CETP expression levels in adipose, kidney and retina tissues, respectively between the T2D mice and control mice. The A-alleles of the promoter SNPs, i.e., rs3764261, rs4783961, rs1800775 were found to be independently associated with 2.86, 1.71, and 0.91 mg/dL increases in HDL-C per allele, respectively. Also, the A-allele of rs4783961 was significantly associated with a reduced T2D risk [odds ratio (OR), 0.82; 95% confidence interval (CI), 0.71-0.96)] and the A-allele of rs1800775 was significantly related to a lowered DKD risk (OR, 0.78; 95% CI, 0.64-0.96), respectively. In addition, the authors applied a T2D mouse model to determine the expression levels of CETP in adipose, kidney, and retinal tissues, and found that CETP expression was significantly reduced in the T2D mice kidney tissue than that in the control mice (T2D mice, Mean: 0.16, SD: 0.01 vs. control mice, Mean: 0.21, SD: 0.02; p=0.02).

Major Comments

This study applied both case-control design and a T2D mouse model to focus on studying molecular effects of genetic variants of CETP on blood lipids levels, and risks of T2R, DKD, and DR, and whether CETP expression is altered in T2D mice adipose, kidney and retina tissue. I have the following major comments.

(1) Page 1, lines 27-28, the authors stated that
"the A-allele of rs4783961 the A-allele of rs4783961 and rs1800775 were significantly associated with a reduced risk of T2D [odds ratio (OR), 0.82; 95% confidence interval (CI), 0.71-0.96)] and DKD (OR, 0.78; 95% CI, 0.64-0.96), respectively."
However, the above expression is confusing, and to make it clearer for the reader audience, could be revised to
"the A-allele of rs4783961 was significantly associated with a reduced T2D risk [odds ratio (OR), 0.82; 95% confidence interval (CI), 0.71-0.96)] and the A-allele of rs1800775 was significantly related to a lowered DKD risk (OR, 0.78; 95% CI, 0.64-0.96), respectively."

(2) Page 3, line 91, the authors stated that
"SNPs—rs3764261 (-2568A/C), rs4783961 (-998A/C) and rs1800775 (-628A/C)"
However, the rs1800775 SNP in CETP gene promoter region is commonly referred to as the -629 SNP (e.g., Tsai MY, et al., Atherosclerosis. 2008;200:359-67. PMID: 18243217; Hishida A, et al.; Lipids Health Dis. 2014;13:162. PMID: 25311932), and therefore, could be revised to
"SNPs—rs3764261 (-2568A/C), rs4783961 (-998A/C) and rs1800775 (-629A/C)"

(3) Page 2 line 67, the authors stated that "The study involved 1640 unrelated individuals with T2D (age >20 years)"
The authors shall provide the age range, rather than just ">20 years". Similarly, Page 2 lines 83, in the statement "The non-diabetic controls consisted of 1383 subjects who were randomly selected", the authors shall provide the age range for these control subjects also.

(4) Page 2, line 72, the authors stated "Of this group, 367 of 1325 T2D subjects were diagnosed with DKD" and also Page 4, lines 151-152, the authors stated "Approximately 27.7% (367/1325) of the T2D 151 patients had DKD", then it raise the concern whether the sample size of T2D subjects shall be 1325, but in Table and Table 4, the authors clearly indicated that the T2D group consisted of 1640 T2D patients, and therefore, at Page 2, line 72, the authors shall clarify how the number "1325" was derived, and in their data analyses, they considered all the 1640 T2D patients in the T2D group, rather this restricted subset.

(5) Page 3, lines 97-98, the authors stated that "Additionally, SNPs that deviated from the Hardy-Weinberg equilibrium were excluded."
What specific Hardy-Weinberg equilibrium (HWE) chi-square p-value threshold was applied for defining the deviation from HWE? Please clarity. In addition, which specific software program was used for testing deviation from Hardy-Weinberg equilibrium (HWE) for each CETP gene SNP (typically HWE is tested based on 1-degree-of-freedum chi-square test)? Please clarify by providing the computer software name and version information.

(6) Page 9, lines 208-210, the authors stated that
"In particular, SNP rs3764261 was associated with an increase 2.86 mg/dL of HDL-C per A-allele in the non-diabetic controls. Although the A-allele increased the HDL-C levels, no significant association was observed with the risk of T2D, or DKD, or DR in our cohort."
This statement is not a sufficient synopsis of the major findings of the study, and shall be revised to
"In particular, SNP rs3764261 was associated with an increase of 2.86 mg/dL of HDL-C per A-allele (p = 1.00E-6), whereas A-alleles of rs4783961 and rs1800775 were associated with increases of 1.71 mg/dL and 0.91 mg/dL of HDL-C per A-allele, respectively at nominal significance level (p = 0.001 and 0.045, respectively),in the non-diabetic controls.  We also observed that at nominal significance level, the A-allele of rs4783961 was associated with a decreased risk of T2D (OR=0.82; 95% CI, 0.71-0.96) and the A-allele of rs1800775 was associated a reduced risk of DKD (OR, 0.78; 95% CI, 0.64-0.96), respectively. However, these results were not adjusted for multiple testing, and shall be confirmed in other larger, independent studies."

(7) A major limitation of the study is the selection of control subjects. As stated by the authors on Page 2, lines 83-85, "The non-diabetic controls consisted of 1383 subjects who were randomly selected from the Taiwan Biobank (TWB) [20], a national large-scale data source with genetic and demographic data from Taiwanese individuals between the ages of 30 and 70 years", however, the T2D patients were recruited from a local region at the China Medical University Hospital (CMUH) at Taichung, Taiwan. As shown in Table 1 (Page 4), the control subjects were significantly younger than T2D subjects (46.0 +/- 10.6 vs 62.0 +/- 11.3; p < 0.001), and had significantly lower BMI (23.9 +/- 3.5 vs 25.4 +/- 4.0; p < 0.001). The non-diabetic controls could be more appropriately selected from the same geographical region where the T2D patients were recruited (for example, from Taichung) and would be better to be randomly selected from the local population with a age range comparable to that of T2D patients, rather than the entire Taiwan, and this point shall be clearly mentioned in the "Discussion" section. Further, In Table 4 (Page 7), In "T2D vs. Non-diabetic controls OR (95% CI) p-value2" column, the logistic regression analysis adjusted for age and gender, and it would be more convincing to have adjusted for age, gender and bmi as covariates in the multivariate regression model, and the authors shall perform such logistic regression analyses and compare with those results obtained only with adjustment for age and gender. Further, in Table 4’s "T2D with DKD vs. without DKD OR (95% CI) p-value3" column, the logistic regression analysis adjusted for age, gender, and HbA1c, and it would be more convincing to have adjusted for age, gender, bmi and HbA1c as covariates in the multivariate regression model, and the authors shall perform such logistic regression analyses and compare with those results obtained only with adjustment for age, gender, and HbA1c. The authors shall revise
Page 9, line 224 "The limitation of the present study as we did not measure CETP activity" to
"There are several limitations of the present study. First, we did not measure CETP activity", and then, shall provide a succinct discussion on the above-mentioned comparisons in the "Discussion" section.

(8) As shown in the study of Li TY, et al., Am J Clin Nutr. 2007;86:1524-9, PMID: 17991668, there could be a gene-diet interaction between the CETP genetic polymorphism and dietary fat intake on plasma HDL-C level, and because this study did not measure dietary fat intake levels in T2D patients and non-diabetic controls, there could be potential confounding of dietary fat intake, which shall be mentioned as one of the several limitations of the study in the "Discussion" section.

Minor Comments

In Table 1 (Page 4), the authors shall put "--" for "P value" column for "Non-DR" subcategory of "Diabetic retinopathy severity scales".

In Figure 1 (b) (Page 8), the authors are suggested to provide the specific p-value of Student's t-test i.e., comparing control mice versus T2D mice for retina tissue.

There are several typographical and grammatical errors that should be corrected, which are shown in the following:

Page 2, lines 49-51,
"Genetic variations in CETP have been studied; while these variants reduce CETP activity, and are associated with the reduced risk of cardiovascular disease [9-13]."
could be changed to
"Genetic variations in CETP have been studied; while these variants are associated with reduced CETP activities, and are associated with the reduced risks of cardiovascular disease [9-13]."

Page 2, line 56,
"a few reports have examined the association of CETP variants with microvascular complications"
could be changed to
"a few reports have examined the associations of CETP variants with microvascular complications"

Page 2, line 58,
"we evaluated the effect of three single nucleotide polymorphisms (SNPs)"
could be changed to
"we evaluated the respective effects of three single nucleotide polymorphisms (SNPs)"

Page 2, line 58,
"we evaluated the effect of three single nucleotide polymorphisms (SNPs)"
could be changed to
"we evaluated the effects of three single nucleotide polymorphisms (SNPs)"

Page 2, lines 76-77,
"for grading the severity of clinical diabetic retinopathy [19]. Among T2D group, 1063 subjects underwent the ophthalmologic examination"
could be changed to
"for grading the severity of clinical DR [19]. Among T2D group patients, 1063 subjects underwent the ophthalmologic examinations"

Page 2, line 81,
"body mass index (BMI), and hemoglobin A1C (HbA1C)"
could be changed to
"body mass index (BMI), and glycated hemoglobin A1C (HbA1C)"

Page 9, "4. Discussion" section, lines 212-213,
"were associated with decreased T2D and DKD 212 risk in our cohort. The increased HDL-C levels may decrease the risk of T2D and DKD."
could be changed to
"were associated with decreased T2D and DKD 212 risks in our cohort. The increased HDL-C levels may decrease the risks of T2D and DKD."

Round 2

Reviewer 2 Report

The current version of the manuscript is considerably improved. I have no further comments.

Reviewer 3 Report

In revised manuscript, the authors appear to have addressed all major and minor comments adequately